# [Re] Boosting Monocular Depth Estimation Models to High-Resolution via Content-Adaptive Multi-Resolution Merging

## Reproducibility Summary

**Scope of Reproducibility**

The authors propose a method to improve monocular depth estimations on multi-megapixel images with existing depth estimation models by merging estimates from lower and higher resolutions. Low-resolution estimates have better structural consistency and lack details, whereas higher resolution images have more details but produce artifacts. Those estimations are merged to an improved base estimate using an image-to-image translation network [1]. The base estimate is further enhanced with local boosting. We aim to reproduce the desired effects on low and high resolution depth maps and verify that merging and boosting improve the accuracy of the final estimate by using the same as well as other depth estimation models and data as the authors.

**Methodology**

We used the code provided by the authors as a baseline and modified it to run the whole pipeline on various depth estimation models. For our benchmark experiments, we considered the same IBIMS-1 [2] dataset as well as another higher resolution dataset not used by the authors, namely DIODE [3]. Our experiments were performed on Google Colab GPU instances (Nvidia Tesla K80) and each benchmark test took several hours, depending on model and dataset.

**Results**

Using the author's code, the pre-trained weights on all models, and the same IBIMS-1 [2] dataset, our error metrics were significantly lower, even though we could still see that the proposed method does indeed improve the overall accuracy metrics of higher resolution depth estimations. The authors then provided us with an updated evaluation method, removing some values from the IBIMS-1 [2] ground truth data, which affected the normalization step. With this unmentioned data processing step, the metrics matched the paper almost exactly. We were additionally able to show the improvements visually as well as quantitatively with other models and other data than the originally used ones.

**What was easy**

The provided code from the authors is simple to navigate and to understand. We did not see any major contradictions to the published paper. Moreover, we were able to easily extend the whole pipeline to run other depth estimation models.

**What was difficult**

The proposed method makes use of multiple inferences per image and with every image having up to 150 patches that need to be estimated and merged into the base estimate, The whole process may therefore be very time-consuming and is computationally expensive overall. Without GPU instances it is not recommended to run the whole pipeline at all.

**Communication with original authors**

The authors provided us with an updated evaluation script, which includes using a guided filter for the IBIMS-1 [2] ground truth data to remove some values and to achieve the same metrics.

# 1 Scope of reproducibility

This report builds on the existing research of inferring depth from images. Current attempts to infer depth for higher resolutions have been impractical for real-world applications due to the lack of fine-grained details from the original estimate or major inconsistencies in the estimated structure. This paper observes the correlation between the input resolution and the resulting depth estimation from cutting-edge neural network models such as MiDaS [4] and SGR [5]. The proposed method builds on the analysis that consistent scene structure, as well as high-frequency details, affect the accuracy of depth estimations. With existing depth estimation models on lower resolution inputs, one can observe more consistent scene structure and depth distribution over the scene with missing high-frequency details. On the other hand, higher resolution input estimates have more details while losing consistency and producing artifacts. The reason for that is the receptive field size, which depends on the architecture of the depth estimation model, can not capture depth if the window corresponds to a relatively small area in the scene not containing depth cues. The paper builds on two factors: creating more accurate depth maps by estimating and merging depth for variable input resolutions and identifying key areas with more depth cues, approximated by edges, for enhanced details via local boosting. We categorized the claims of the paper as follows:

- Claim 1: Input image resolution affects the performance of depth estimation models. Low input resolution estimations have a consistent scene structure but lack fine-grained details. High input resolution estimations contain details but structures are broken and artifacts are occurring.

- Claim 2: Merging low and high resolution estimates with an image-to-image translation network [1], denoted as *double estimation*, gives better performance compared to each individual estimate.

- Claim 3: selecting local patches and boosting them improves performance even more by preserving high-resolution details in key areas containing depth cues.

- Claim 4: The double estimation method with local boosting generalizes to better performance across existing depth estimation models and high-resolution input data.

Claim 1 and Claim 2 were verified qualitatively through observing different high-resolution sample images estimated using low and high resolution inputs. To verify Claim 3, we made use of depth models and investigate the detail enhancement of high-resolution images compared to the base estimation. Our reproducibility focus was set on Claim 4, which is the main research contribution of this paper. A generalized method for improved high-resolution estimates, which was verified quantitatively by evaluating the same as well as other depth inferring models and datasets than the authors.

# 2 Methodology

The below subsections will further elaborate on the individual parts of the proposed method and what resources we used to verify claims made by the authors, including which code, models, and datasets.

## 2.1 Method and model descriptions

The authors claim to improve existing depth estimation models and therefore we will compare the results achieved with 4 different models: SGR [5], MiDaS v2 [4], LeReS [6], and MiDaS v3 [7].
We denote the estimations with those models without any modifications as *the original* estimate, which is usually a low-resolution estimate depending on the receptive field size. Each model is based on different architectures and vary on computational effort, specifically, MiDaS v3 [7] is based on recently proposed Vision Transformers [8] and comes in two different pre-trained sizes, *DPT-hybrid* and *DPT-large*. Due to limited GPU resources, we used the smaller DPT-hybrid model and we will refer to it simply as MiDaS v3. All the models used were pre-trained with weights given by the original authors of each model. We aim to verify the overall claim of improving existing estimates on higher resolutions using multiple inferences on different input resolutions and patches through the same model, which are merged to result in a *final* estimate.

The whole process of improving the depth estimate of existing models on higher resolutions consists of several steps:

1. Determine the optimal high resolution for a given image ($R_{20}$)
2. Use the model to estimate depth for low and high resolution
3. Merge both estimates using the merging network to create the so-called *double estimate* as base
4. Generate relevant patches
5. Iterate through patches, double estimate each patch, and merge into the base estimate

This is also visualized in Figure 2 with intermediate results in the pipeline.

Regarding the optimal resolution search, the authors state that the structural inconsistency is due to the pixels that do not receive any contextual cues in their receptive field, which we also observed in our experiments, see Figure 1. The receptive field refers to the area around a pixel that contributes to the estimation there and is determined by the network structure and training resolution. The resolution that every pixel is at most a half receptive field size away from context edges is called $R_0$, and it is the resolution that ensures the structural consistency. On the other hand, the resolution where 20 percent of pixels in the image do not receive any contextual cues is denoted as $R_{20}$. The distribution of contextual cues in the image is approximated by an edge map. $R_{20}$ is assumed to be the optimal high resolution for the method, however, in the code it is limited by a threshold for the reason of limited GPU memory.

To generate relevant patches, the image is first tiled with a size equal to the receptive field size and a 1/3 overlap, where each tile is a candidate patch. Patches with low gradients in the image, derived by the edge map, are discarded whereas patches that have more contextual cue density are enlarged until the edge of the original image is reached. Then for each patch, a double estimation (low and high resolution depth merged estimation) is computed, which is again merged into the base estimate with the same merging network as before.

A crucial component in the pipeline of generating high-resolution depth maps is the merging network (called `pix2pix` in the code), which has the task of merging low and high resolution estimates as well as patches into the base estimate. Based on the standard architecture of an image-to-image translation network [1], it was trained by the authors using paired data generated from patches of the Middlebury 2014 [9] and IBIMS-1 [2] datasets. We did not retrain the merging network, nevertheless, we examined the provided code and the detailed instructions in the repository on preparing the data for retraining it from scratch.

## 2.2 Datasets

We evaluated the algorithm with the two datasets IBIMS-1 [2] and DIODE [3]. However, since we were limited in available processing power we only used 100 samples of the DIODE [3] dataset, 5 samples from each scene of the evaluation dataset consisting of outdoor and indoor scenes. The IBIMS-1 [2] dataset was providing RGB depth images, however, the DIODE [3] dataset only provided values as Numpy array files (`.npy` extension). Therefore the depth data was pre-processed by converting the depth files to `png` files such that the evaluation method, provided in the repository and written in Matlab, does not need to be changed. The evaluation code was only modified after receiving the hint from the authors that they used a guided filter to remove values influencing the normalization step in the IBIMS-1 [2] dataset.

IBIMS-1: 100 samples, link: https://www.asg.ed.tum.de/lmf/ibims1/
DIODE: 100 samples, selected and pre-processed as described above, link: https://diode-dataset.org/

## 2.3 Hyperparameters

Hyperparameter selection or optimization was not required since we used pre-trained models provided by the authors of each component to understand and report intermediate results of the pipeline.

## 2.4 Experimental setup and code

The overall setup is well explained in the author's repository, which also provides a demo notebook to run SGR [5], MiDaS v2 [4], and LeRes [6]. The download links to the weights of each network are also included. The benefit of using this method is that any depth estimation model can easily be added and run through the pipeline since the integration is simple and clear. The only thing one needs to adjust to include their own depth estimation model is adding the loading of the model and writing the single estimate method with a resolution size as an argument for resizing

inputs and changing the command line arguments accordingly. That is exactly what we did for extending the pipeline to include MiDaS v3 [7], with some code portions taken from the MiDaS v3 [7] code repository.

The command-line arguments provide an interface to simplify running the whole pipeline, such as selecting the depth estimation model, saving patches as well as the base estimates for each picture. Furthermore, we added a command-line argument to save the low and high resolution estimates before they were used to create the base estimate. Since we had to run our experiments on Google Colab, where sessions might lose connection due to limited runtimes, we added slight modifications such as a simple check to skip images that were already estimated and finished.

The metrics to measure how well our results performed are the same as the ones in the paper:

- ORD: ordinal error
- $D^3R$: depth discontinuity disagreement ratio
- RMSE: Root Mean Square Error
- $\delta_{1.25}$: percentage of pixel with $\delta = \max \frac{z_i'}{z_i}, \frac{z_i}{z_i} > 1.25$

The metric $D^3R$ specifically was introduced by the authors to emphasize more on details such as edges. It is based on ORD with the addition of (scaled) superpixel. We also used the same Matlab script as the authors to generate the evaluation values. The scale for the superpixel in $D^3R$ was set as the comments in the script describe for MiDaS v2 [4] at 1.0 and for DIODE [3] we set it to 0.5. Scaling was introduced by the authors to reduce the runtime for evaluation, which took around 10 minutes for a 100 image dataset, such as IBIMS-1 [2]. Another note is that LeReS [6] was estimating depth, while SGR [5], Midas v2 ,[4] and MiDaS v3 [7] estimate inverse depth (disparity). Therefore the evaluation needs to be adjusted accordingly, which we did by swapping depth and disparity when evaluating LeReS [6].

Running the original depth estimation models for single estimates was either done by using adjusted demo scripts of each model or the command line argument which we added, since the low-resolution estimate, i.e. receptive field size as input resolution, corresponds to the original output.

The author's code repository is available at:
https://github.com/compphoto/BoostingMonocularDepth
We used a fork of their repository for our experiments and modifications, most of them being in the `run.py` file or the `our_experiments` directory within the repository, available at:
https://anonymous.4open.science/r/BoostingMonocularDepth-BDD1

## 2.5 Computational requirements

We adjusted the author's code to make it executable on CPU-only machines. However, it is very time demanding to run such large models such as LeReS [6] locally on a CPU instance and memory was an additional problem. Therefore, we used Google Colab to obtain our results, which is also not recommended since the runtimes are limited and it might take several days to have all the results. Having a stronger Nvidia GPU available will presumably result in more reasonable runtimes of a few hours for a whole dataset. Our runtimes are based on our estimates of executing the whole dataset in one run since we had to execute the datasets partially over multiple days considering Colab time-outs using the Nvidia K80 GPU. We can say that the method is not very accessible to users without a GPU, especially the local boosting step. A plain double estimation without the local boosting would result in more reasonable runtimes, performing moderately well compared to a single estimation.

|            | IBIMS-1 [2] | DIODE [3] |
|------------|-------------|-----------|
| SGR [5]    | 3h          | 4h        |
| MiDaS v2 [4] | 6h        | 8h        |
| LeReS [6]  | 10h         | -         |
| MiDaS v3 [7] | 12h       | 15h       |

Table 1: Estimated runtimes for each dataset evaluation and model on a single Nvidia K80 GPU.

The evaluation script in Matlab takes approximately 10 minutes for both datasets of 100 images on a local CPU instance, with DIODE [3] superpixel being scaled by 0.5.

## 3    Results

In this section we will present our results achieved, observing the effects described on lower and higher resolution estimates as well as several dataset evaluations from the paper and beyond.

### 3.1    Results reproducing original paper

#### 3.1.1    Result 1

To verify the first two claims, as we categorized in Section 1, we tested all our analyzed depth estimation models to generate low and high resolution images as well as computing the base estimate after merging them. One of our sample outputs is shown in Figure 1 using the MiDaS v2 [4] model.

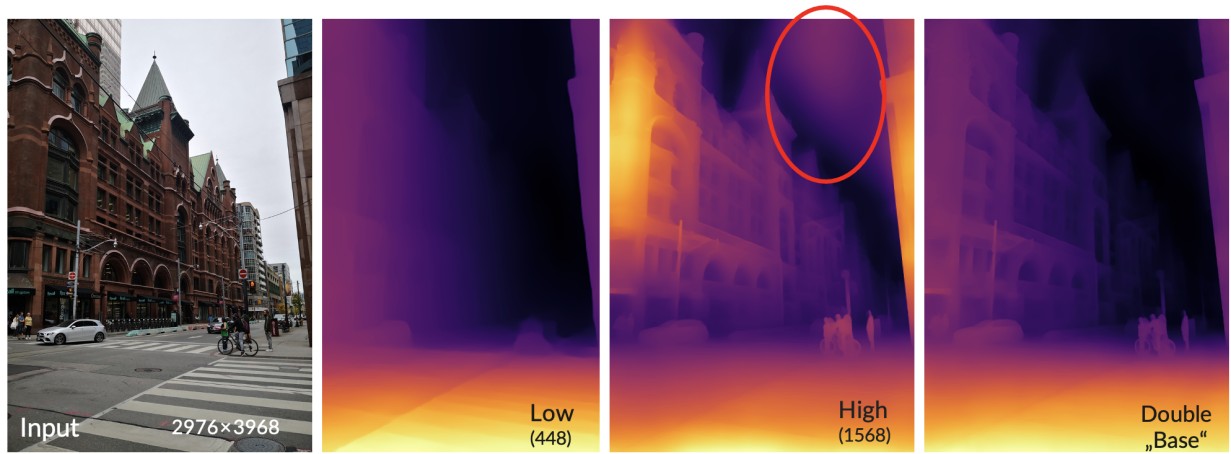

Figure 1: Example of running a high resolution (12 megapixels) image depth estimation with MiDaS v2 [4] with lower and higher ($R_2 0$) resolution input size. Artefacts in the high resolution estimate resulting from insufficient depth cues are marked red. On the right we can see the result after merging the low and high resolution image.

#### 3.1.2    Result 2

We further visually observed in Figure 2 the improvements of using local boosting to question Claim 3 in Section 1 using LeReS [6] as a depth estimation model.

#### 3.1.3    Result 3

Claim 4 from Section1 was set to be our main focus since the method should generalize to any depth estimation model on higher resolution inputs. To have a metric comparison to the paper published by the authors, we estimated the same dataset IBIMS-1 [2] with the same depth estimation models MiDaS v2 [4] and SGR [5]. Our initial results resemble the metrics achieved before removing some values of the ground truth data as suggested by the authors. The *filtered* values correspond to the results with the author's addition.

Our initial and filtered results along with the values published by the authors are presented in Table 2.

### 3.2    Results beyond original paper

Our additional experiments and results use other networks and datasets to emphasize more on Claim 4, generalizing to other depth estimation models and high-resolution inputs.

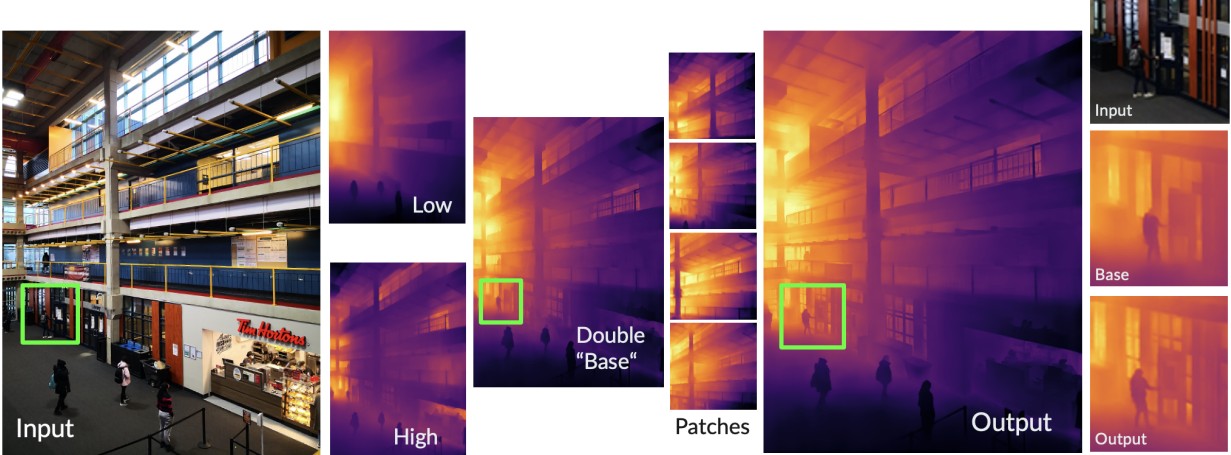

Figure 2: Example of running a high resolution (12 megapixels) image through the method with intermediate results, such as low and high resolution estimates as well as patches using the depth estimation model LeReS [6]. The area marked in a green box was magnified on the right side for better visualization of the local boosting effect.

| | IBIMS-1 [2] | | | | | | | |
|---|---|---|---|---|---|---|---|---|
| | MiDaS v2 [4] | | | | SGR [5] | | | |
| | ORD | D³R | RMSE | $\delta_{1.25}$ | ORD | D³R | RMSE | $\delta_{1.25}$ |
| author's original method | 0.4002 | 0.3698 | **0.1596** | **0.6345** | 0.5555 | 0.4736 | **0.1956** | 0.7513 |
| author's final result | **0.3938** | **0.3222** | 0.1598 | 0.6390 | **0.5538** | **0.4671** | 0.1965 | **0.7460** |
| Our original method (filtered) | 0.4015 | 0.3698 | **0.1596** | **0.6345** | 0.5488 | 0.4814 | **0.1963** | 0.7532 |
| Our final result (filtered) | **0.4011** | **0.3215** | 0.1600 | 0.6365 | **0.5484** | **0.4694** | 0.1964 | **0.7457** |
| Our original method (initial) | 0.2687 | 0.3429 | 0.1821 | 0.6147 | **0.3971** | 0.4247 | 0.2531 | 0.7914 |
| Our final result (initial) | **0.2644** | **0.2552** | **0.1792** | **0.6073** | 0.3980 | **0.3708** | **0.2491** | **0.7791** |

Table 2: Quantitative evaluation and comparison of our filtered and initial results to the ones obtained from the paper using the same dataset and models.

### 3.2.1 Additional Result 1

The first additional result we provide is running again the same dataset with the same metrics as a benchmark, however, using two different depth estimation models, namely MiDaS v3 [7] and LeReS [6]. The results can be found in Table 3. Due to lacking memory and time some samples were not able to run on our hardware. More than half of the dataset was evaluated for the original and final estimations to have a fair comparison with the results in Table 3.

| | IBIMS-1 [2] | | | | | | | |
|---|---|---|---|---|---|---|---|---|
| | MiDaS v3 [7] | | | | LeReS [6] | | | |
| | ORD | D³R | RMSE | $\delta_{1.25}$ | ORD | D³R | RMSE | $\delta_{1.25}$ |
| Our original method (filtered) | 0.4005 | 0.3286 | 0.1519 | 0.6414 | **0.3403** | **0.2951** | **0.1527** | **0.6677** |
| Our final result (filtered) | **0.3911** | **0.3121** | 0.1519 | **0.6370** | 0.3714 | 0.3178 | 0.1547 | 0.6844 |
| Our original method (initial) | **0.1992** | 0.2387 | 0.1509 | **0.5741** | 0.1943 | 0.2626 | 0.1117 | **0.4630** |
| Our final result (initial) | 0.2090 | **0.2144** | **0.1501** | 0.5866 | **0.1510** | **0.2205** | **0.0919** | 0.6918 |

Table 3: Quantitative evaluation and comparison of our filtered and initial results using the same dataset but different depth estimation models than the authors.

## 3.2.2 Additional Result 2

To mitigate the doubt of IBIMS-1 [2] and Middlebury 2014 [9] datasets being specific examples of their method performing well since they also trained the merging network with those two datasets, which would contradict the claim of generalization, we benchmarked their method against the DIODE [3] high-resolution depth dataset in Table 4 with three depth estimation models.

| | DIODE [3] | | | | | | | | | | | |
| | MiDaS [4] | | | | SGR [5] | | | | MiDaS v3 [7] | | | |
| | ORD | $D^3R$ | RMSE | $\delta_{1.25}$ | ORD | $D^3R$ | RMSE | $\delta_{1.25}$ | ORD | $D^3R$ | RMSE | $\delta_{1.25}$ |
|---|---|---|---|---|---|---|---|---|---|---|---|---|
| Our original method | 0.3692 | 0.6617 | 0.2960 | **0.8852** | 0.5025 | 0.6922 | **0.3361** | 0.9162 | 0.2804 | 0.6468 | 0.2841 | **0.8658** |
| Our final result | **0.3469** | **0.6327** | **0.2936** | 0.8876 | **0.4983** | **0.6787** | 0.3381 | **0.9146** | **0.2657** | **0.6209** | **0.2829** | 0.8706 |

Table 4: Quantitative evaluation and comparison using a different dataset than the author's across three different depth estimation models.

# 4 Discussion

As our results show, we were able to support all claims with our experiments. The first claim of observing better structural consistency but lacking details on lower resolution estimates, while higher resolutions have more details but produce artifacts were clearly visible in examples with areas missing depth cues (low gradients). In our example from Figure 1 we could observe the effect in the sky of the image being relatively consistent without any edges. By merging both of those estimates and creating the base estimate, we leverage the advantages of both input resolutions which were also visible in the same Figure 1. With local boosting, by estimating the depth of patches, additional details were distinctly visible in the results, such as shown in Figure 2. Lastly, in our benchmark results, running the same experiment with the same pre-trained model and dataset has initially proven to result in vastly different values for our error metrics. We were not able to explain why there was a difference, however, our results do not contradict the overall claim of improved depth estimation using their method. After approaching the authors about those differences, we also added the provided data processing step of removing some values from the IBIMS-1 [2] dataset using a guided filter. This data processing step for evaluation was not highlighted or mentioned in the paper published and we did not fully understand its necessity since we did not run into any problems with the min-max-normalization. In fact, our initial results for the IBIMS-1 [2] dataset were bringing even more attention to the improvement of using their method, especially the $D^3R$ error among others decreased significantly compared to the filtered results. Even with other models in Table 3 such as LeReS [6], the improvement was more obvious than using the filtered data for evaluation.

In general, using other models such as MiDaS v3 [7] or LeRes [6] did not contradict their proposed method since they improved the quantitative results on IBIMS-1 [2], at least in the initial comparison while the filtered data resulted in worse improvements. Similar performance gains were observed using the DIODE [3] dataset and therefore we can conclude that we were able to reproduce improved higher-resolution depth estimations using their proposed method.

## 4.1 What was easy

The provided code by the authors is clearly written, simple to navigate and understand. They also implemented a command-line interface to run with different settings, such as choosing the depth estimation model. Also, their documentation in the `README.md` file and the additional resources help to understand the concepts of their method as well as navigate to repositories of each model, such as MiDaS v2 [4].

With their code as a baseline, the effort to run your own depth models are minimal i.e. by extending the command line interface to load the model, adding the model-loading code, and adding the method to process a single estimate through the new model in variable size, which we did for MiDaS v3 [7].

## 4.2 What was difficult

Depending on the image resolution and high-frequency details, the number of patches reaches up to 150 in our experiments and with every patch, a double estimation (low and high resolution estimate followed by merging) needs

to be computed, which are many inferences of the depth estimation model compared to the single estimation of the original method. Since we were limited in available computing resources, running all our benchmarks on free Google Colab GPU instances, the benchmarks took several days to complete, which is relatively long for just evaluating a method. Despite our modifications to run the code on CPU only, it is not recommended to run the code on CPU-only machines, except for single images which may take up to an hour depending on the resolution. With more computing resources and time we would have tested more quantitative evaluations of intermediate results, focusing on individual contributions of the pipeline to the highly improved accuracy of the high-resolution depth estimation.

### 4.3 Communication with original author's

After inquiring the authors for the reason we initially achieved different values for the error metrics, they provided us with an updated version of the evaluation script. They used a guided filter on the ground truth IBIMS-1 [2] data to detect and remove points affecting the min-max-normalization. In the paper itself, this fact was not mentioned anywhere.

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
