# OpenReview forum: "[Re] Boosting Monocular Depth Estimation Models to High-Resolution via Content-Adaptive Multi-Resolution Merging"
_ML_Reproducibility_Challenge/2021/Fall — Reject_

### Official Review · Reviewer_VA5r · 2022-03-07
**lack of hyper-parameter studies**

**Rating:** 6
**Confidence:** 4

**Review:**

This paper reproduced the original paper that proposed merging the low-resolution and high-resolution depth estimations with an image-to-image translation network to boost the depth estimation map.

This paper is based on the code provided by the original paper's authors.

The authors further compare the results based on 4 different depth estimation models.
The main discovery of this paper is that the original paper uses a specifically processed ground truth depth map in the evaluation, i.e., some undesired components in the ground truth depth maps are removed.



The authors only studied the merging process of the proposed method by using the pretrained merging model provided by the original authors.
This paper does not study the reproducibility of the training of the “image-to-image translation network”. Thus the influences of the hyper-parameters are not studied.
This limits the significance of this paper.

---

### Official Review · Reviewer_Mx7E · 2022-03-27

**Rating:** 5
**Confidence:** 3

**Review:**

Pros:
* Mostly clear and effectively communicated, no typographic problems
* Outline of original paper claims and reproducibility pipeline is good
* Additional benchmarks on different datasets
* Readable codebase is provided

Cons:
* Same metrics, models, and parameters as the original paper
* There is nothing wrong with using pre-trained weights, but the evaluations of the datasets are severely limited by the lack of compute

Summary of review:
Overall, this is a well written report that has the potential for some interesting evaluations of the original work in new settings. However, it is hindered by a lack of experiments/evaluations (which the authors attribute to the lack of compute). This lack of compute is an understandable problem, but it might be better to select a paper which requires less compute to work with. Some expansions that might be nice to see in the paper include things that would require substantial compute (such evaluating the model with different training hyperparameters) and a more moderate amount (such as full evaluation on a large suite of benchmarks). Although the results presented are interesting, their limitations preclude me from recommending the paper.

---

### Meta-Review · Area_Chair_XSbD · 2022-04-08

**Recommendation:** Reject
**Confidence:** 4

**Metareview:**

While this is a good study, the experiments are reproduced by using the pre-trained weights which makes it hard to evaluate the effectiveness of the reproducibility study.

---

### Decision · Program_Chairs · 2022-04-09

Reject